# Confronting SARS-CoV-2 Infection: Patients’ Experience in the First Pandemic Wave—Cross-Sectional Study

**DOI:** 10.3390/ijerph191912743

**Published:** 2022-10-05

**Authors:** Maja Socan, Vanja Ida Erčulj

**Affiliations:** 1National Institute of Public Health, 1000 Ljubljana, Slovenia; 2Faculty of Criminal Justice and Security, University of Maribor, 2000 Maribor, Slovenia

**Keywords:** COVID-19, first pandemic wave, isolation, information, social support, feelings, stigma

## Abstract

The aim of the study was to investigate the real-life experience of persons infected with SARS-CoV-2 in Slovenia in the first pandemic wave and how the buffering effect of social and informational support affected negative feelings. We used a self-administrated questionnaire. There were 1182 eligible notified cases with the response rate 64.9%. At least 62% of responders were able to follow the isolation rules, while 21.1% did not or could not organize their living separately from other household members. The main providers during the isolation period were close family members. The most prevalent emotion in our study was worry (70.3%) and fear (37.6%). Worry and fear during the illness were less probable for men than women, but more probable for older patients. Participants with strong emotional support had lower odds of being sad. Those who were exposed to a larger number of sources of information had higher odds of being worried. Those patients who used a higher number of more credible sources of information had higher odds of being afraid during illness. Pets did not play a special role in psychological well-being. The role of the media and public health communications should be explored further to achieve an improved response.

## 1. Introduction

Pandemics occur unexpectedly and have a pronounced global effect. The last pandemic caused by SARS-CoV-2 virus started at the end of 2019 in China [1]. At the beginning of the COVID-19 pandemic, there was insufficient information available about transmission, the natural course of the disease, and population groups at the highest risk. There was no vaccine available and limited knowledge about effective public health measures to reduce the number of new cases [2]. Unfavorable reports from countries that were the first to face larger outbreaks of COVID-19, characterized by high morbidity and mortality, led to the conclusion that very strict public health measures were needed to control the evolving situation [3,4]. Concerns about the anticipated high health system overload required an immediate response by decision-makers. Many countries introduced stringent non-pharmaceutical measures (NFM) never used before, e.g., cessation of public transport, closure of shops and services, school closures and distance learning, work from home, etc. [5]. 

The quality of life drastically changed in a short period of time after the beginning of the pandemic [6,7]. The lockdown reduced face-to-face social interactions with many negative effects on physical and mental well-being [8,9]. Mobility was restricted, and there was reduced engagement in human communication, including more disconnection from both formal as well as informal support [10,11,12]. Due to the “stay at home” recommendation, which was intended to reduce uncontrolled transmission of SARS-CoV-2, there was more time spent indoors, eating behavior changed, alcohol consumption increased and physical activity decreased [13,14]. There was an important shift towards mostly unhealthy lifestyle patterns [15]. In general, the accessibility of healthcare decreased as medical facilities around the world were not prepared for the enormous challenges posed by the growing pandemic [16]. 

The COVID-19 pandemic has led to economic and psychological exhaustion for people globally. Many studies have investigated mental health in the community during the first pandemic waves. Feelings of loneliness, worry and stress levels rose with increased rates of reported anxiety and depression symptoms [17]. Risk factors associated with distress measures were being younger, female, unemployed, having co-morbidity, and being frequently exposed to social media/news regarding COVID-19 [17]. Perceived and received social support had a buffering effect and was associated with higher psychological health, and insulated individuals from psychological problems caused by the COVID-19 pandemic situation [18,19]. Pets were also one of the important sources of comfort during COVID-19, especially during confinement [20]. 

The COVID-19 pandemic was intensively covered by traditional media and also used by governments to convey important information and recommendations to the general public and news shared on social media. Too much information, including false or misleading information, in digital and physical environments during the pandemic was observed [21]. Infodemics combined with a changing and unstable environment created a stressful situation in society [22,23]. 

In the first pandemic wave, the number of COVID-19 patients was low in Slovenia due to the NFM being introduced soon after the first case was confirmed. The aim of our study was to investigate the real-life experience of persons infected with SARS-CoV-2 in Slovenia during the first pandemic wave. We explored the emotions at diagnosis and during the course of the disease, the source of information about COVID-19, the role of the media in experiencing the illness and social support provision. The purpose was to study the buffering effect of emotional and informational support on negative feelings during the course of the disease. 

## 2. Materials and Methods

### 2.1. Participants

This cross-sectional study employed a self-administered survey as the data collection tool. Self-administrated questionnaires were sent to SARS-CoV-2 non-institutionalized PCR positive cases notified from 4 March 2020 (first confirmed case in Slovenia) to 31 May 2020 (end of the first pandemic wave). The sample collection process was included in the Appendix A. We excluded residents of long-term care facilities. The questionnaires were not sent to the home addresses of the 23 patients who died due to COVID-19 and were not residents of LTCF. The rest of the patients who died were nursing home residents (86 out of 109 who died up until 1 June 2020, 78.9%). 

The research was conducted in June 2020. The survey was administered via post at home addresses of SARS-CoV-2 notified cases. Participants were informed about the study’s purpose, the identity of the researchers and how the data would be stored in a section at the beginning of the form. 

### 2.2. Assessment Variables

The self-administered survey included demographic, health-related and socio-economic data (age, gender, co-morbidities, level of education, living conditions), the organization of isolation during infectivity period (including procurement of daily necessities) and questions regarding psychosocial health, stigma and information-seeking behaviors. The questionnaire is available in the Appendix A.

### 2.3. Statistical Methods

Categorical variables were described by frequencies and percentages and non-normally distributed continuous variables by medians and interquartile ranges (IQR). 

The association between each emotion at diagnosis and during the illness was tested with a McNemar test. A strong social support variable was formed from receiving social support from three sources, family members and friends, acquaintances and co-workers, and medical staff. If all three sources were present, then the received social support was strong. A separate variable with the number of the most credible main sources of information was formed. These included the Ministry of Health, the National Institute of Public health (NIPH), medical staff and scientific literature. The association between demographic variables, social support, sources of information and worry, fear and sadness during the illness was tested by three multiple logistic regression models. There was no multicollinearity between the predictors (highest variance inflation factor Vif equalled 1.1). All the associations were tested at the level of significance α = 0.05. 

## 3. Results

In first pandemic wave, there were 1182 notified SARS-CoV-2 cases eligible for the study (after excluding LTCF residents and 23 patients who died in the first month after infection). The response rate was 64.9% (767 notified cases responded). The respondents left some questions of the survey unanswered. 

Socio-demographic data and co-morbidities of SARS-CoV-2 notified cases in the first pandemic wave that responded to the survey are summarized in Table 1. There were 418 (55.5%) females in the sample. The age of half of the participants was 49 or less (IQR: 33–59). There were 89 (11.9%) participants with an elementary education, 327 (43.8%) with secondary and 331 (44.3%) with higher (university) education. Half of the respondents (381, 49.7%) had a concomitant chronic disease. 

Out of 756 participants, 503 (66.5%) needed medical advice during the illness, while a bit less than a quarter (23.1%) of respondents that provided answers to the questions were hospitalized (Table 2). Almost two thirds (63.7%) out of 760 respondents still had persistent symptoms after 15 days of illness. In addition, 325 (43.6%) out of 746 were able to carry on with their everyday activities in the same way as before the illness. 

Less than half (44.7%) of participants out of 767 reported having a pet in the household. A total of 463 (62%) isolated themselves from the other members of the household by living in a separated house, floor or at least a room in the house or apartment. Furthermore, 53 SARS-CoV-2 positive persons did not isolate from other household members as the whole family was infected. Almost all (716; 95.9%) respondents reported having strong social support from the closest family members and friends, 572 (78.1%) from acquaintances and co-workers and 615 (84%) from medical staff (Table 3).

The main caregivers providing patients with various supplies and food were the closest family members (52%) or family members living in the same household (41%). For 29% of the respondents, the main caregivers were (also) friends and for 17% their neighbours (Figure 1). 

The prevailing feeling at the time when infection with SARS-CoV-2 was confirmed and during the course of the illness was worry (Table 4). About two thirds of respondents reported that they worried at the beginning and during the course of the illness. 

Figure 2 illustrates the emotional states that were present at the time of diagnosis that still persisted during the time of treatment. There is a statistically significant association between each emotion being present at the beginning (time of diagnosis) and during the illness (*p* < 0.001 for all emotional states). 

Figure 3 illustrates the share of patients using each source of information about COVID-19 and which they considered the most reliable source of information. The three predominant sources were television (63%), medical staff (58%) and the telephone number provided by the National Institute of Public Health (41%). The first two above-mentioned were also indicated as the most reliable sources of information. 

The association between demographic variables, sources of social support and information and each of the main negative feelings (worry, fear and sadness) during the course of the illness was tested by three multiple logistic regression models, and the results are summarized in Table 5. Worry and fear during the illness were less probable for men than women (OR (95% CI): 0.65 (0.44–0.95 and 0.47 (0.32–0.69], respectively), but more probable for older patients (OR (95% CI): 1.04 (1.03–1.05) and 1.02 (1.01–1.03), respectively), when controlling for other predictors in the model. Having a pet was not statistically significantly associated with any of the emotions, while strong social support from all three sources (family members and friends, colleagues and co-workers, and medical staff) was statistically significantly associated with the feeling of sadness during the illness. Patients with strong emotional support had lower odds of being sad (OR (95% CI): 0.48 (0.27–0.88)). Strong social support also alleviated the fear the patients felt, but the association is borderline significant (*p* = 0.067). Sadness was to a lesser extent one of the accompanying feelings during the illness for those patients that were not socially isolated in the process (OR (95% CI): 0.59 (0.37–0.93)), but no statistically significant association was found between social isolation and presence of fear or worry (*p* = 0.446 and *p* = 0.676, respectively).

Patients exposed to a larger number of sources of information had higher odds of being worried (OR (95% CI): 1.26 (1.12–1.43)), but no statistically significant association was found between the number of information sources and being afraid or sad during illness. Those patients who used a higher number of more credible sources of information (Ministry of Health, NIPH, medical staff and scientific literature) had higher odds of being afraid during illness (OR (95% CI): 1.65 (1.22–2.21)). No statistically significant association was found between the number of the most credible sources of information used and feeling worried or sad (*p* = 0.472 and *p* = 0.217, respectively).

## 4. Discussion

### 4.1. Main Findings

In this study, we surveyed the experience of COVID-19 first-wave (from 4 March to 31 May 2020) patients who were not long-term care facility residents. Approximately two thirds completed the questionnaire. There were slightly more female (55%) than male respondents, with a median age of 49 years. The respondents have on average a higher level of education compared to the general Slovenian population according to the official statistics (source: Statistical Office, https://www.stat.si/statweb/en, accessed on 3 September 2022). In the first pandemic wave, at least at the beginning, more cases were related to business and leisure travel abroad. After the importation of SARS-CoV-2, the virus was spread within the socio-economic strata to which initial cases belonged. There were more healthcare workers, teachers, CEOs, business people, etc. in the first wave cohort, which explains the deviation from the general population in educational attainment. Half of the respondents had a chronic disease, with high blood pressure being most often listed. One third of SARS-CoV-2 positives did not consult their primary care physicians, most probably because they were only asymptomatically infected. In the first pandemic wave, the close contacts of patients with COVID-19 were tested to rule out infectivity during and at the end of quarantine and often found to be only asymptomatically infected. Less than half (43.6%) of responders were able to carry on daily activities two weeks after the beginning of illness. Nearly two thirds reported persistent symptoms on the 15th day of the illness. Prolonged convalescence and persistence of symptoms have been well documented [24]. 

Real-life experience showed that COVID-19 patients faced many practical issues while organizing their daily lives according to the rules of isolation and quarantine [25]. In our study, at least 62% of responders were able to follow the isolation rules, but 21.1% did not or could not organize their living separately from other household members—a similar percentage was found in the study from the USA [25]. In some families, all were infected concomitantly, while, in others, the living conditions did not allow self-isolation from other members of the household. The lack of sufficient space and sanitation facilities to comply with recommendations to isolate or quarantine may impede the effective prevention of household spread of COVID-19 or other microbes in future pandemics. 

### 4.2. Social Support

The number of COVID-19 cases in the first pandemic wave in Slovenia was very low. For those who were infected and isolated, it was not difficult to find a provider for procurement and delivery of food and other necessities. Household members, extended family members and friends were willing to help, and, in rare cases, the Red Cross or Civil protection were activated to support everyday living. It would be interesting to repeat the study in the autumn-winter 2021/2022 pandemic waves, when the number of COVID-19 patients was much higher, and learn how families functioned—it might be that previous experience with COVID-19, isolation or quarantine led to better organization for procurement with food and other items, or another possibility is that there was less rigid compliance with isolation rules [26]. 

### 4.3. Feelings

The important finding of our study was that strong social support reduced fear and sadness in patients during illness—in accordance with the theory of the buffering effect of social support [15]. Patients who were not in isolation had fewer feelings of sadness. In our study, male participants had lower odds for negative feelings such as worry and fear with no difference found regarding sadness compared to females. Female gender was reported frequently, but not consistently as risk factors for fear, anxiety and depression during the COVID-19 pandemic [27,28]. Pets did not play a special role in psychological well-being. 

A systematic review of the psychological experience of COVID-19 patients showed that fear was the prevailing emotion, at least at the time of diagnosis. Mandatory isolation and uncertainty regarding the course and outcome of COVID-19 were key sources of fear [29]. Interestingly, the most prevalent emotion in our study was worry (70.3%). Fear was described by 37.6% of participants at the beginning of the illness. The survey did not include detailed questions about the worries that afflicted those infected with SARS-CoV-2, which is one of the limitations of our study. We anticipated that there were two main concerns: the progress of the disease and the danger of infecting their family members, as shown by previous research [29]. One fourth of respondents reported that they felt perplexed and approximately one fifth were angry at the time of diagnosis, which is similar to other first wave studies [29,30]. It appears that many first wave COVID-19 patients did not foresee the possibility of getting infected. An Australian study reported that some COVID-19 patients were shocked when they learned that they were infected as they did not consider themselves to be at any risk [30]. 

### 4.4. Stigma

Stigmatization is defined as social disapproval or negative perception of an individual because of his characteristics or attributes [31]. Stigmatization of those infected with SARS-CoV-2 and even of those who might be infected and their family members has been well described in published literature [32]. The stigma of COVID-19 affected some professional, social, religious groups and national communities. An Indian study from the first months of the pandemic found that two thirds of healthcare workers had at least one stigmatizing experience, and approximately half reported a high impact of stigma on their social life [33]. The COVID-19 pandemic has fueled pre-existing racism in multi-ethnic, multi-cultural and multi-religious countries, and became a convenient excuse for inappropriate responses or even refusing medical assistance [32]. The psychological burden of such stigma strongly influences the individual’s readiness to inform others about infection or to seek healthcare, which in turn fuels the uncontrolled spread of the virus. Destigmatizing COVID-19 patients is a step to more effective control of pandemic [31]. The stigma of COVID-19 is not rare—from one fourth to more than two thirds of COVID-19 patients perceived and experienced stigmatization [34,35,36]. In our study, 22.4% respondents encountered stigma, and 25% had a negative experience but were not sure if it was stigmatizing. A systematic review of the prevalence of stigma in infectious diseases (including COVID-19) found that the prevalence of stigma in participants from low- and middle-income countries was higher compared to high-income countries, but not statistically significantly different [32]. A lower level of education proved to be an important factor for higher prevalence of stigma [32]. Pandemic preparedness plans should use the experience of the COVID-19 pandemic and incorporate this into preventive strategies [37].

### 4.5. Information

An online survey conducted among US adults during spring 2020 using convenient samples showed traditional media sources (television, radio, podcasts or newspapers) were the largest sources of COVID-19 information [38]. The COVID-19 patients included in our study used multiple media to obtain information, with television being used most often. Healthcare workers were the second most frequent and most important source during the illness. New media social networks and websites were used but not with high frequency, as observed in a Chinese study which showed that new media (social media and search engines) were the main channel by which people obtained information on COVID-19 [39]. Mainstream media are able to actively broadcast appropriate information and useful recommendations for disease control (e.g., how to use personal protective equipment properly, respect social distancing, and giving updates on lockdowns and government responses). The media are inclined to sensationalism, which in most cases does not empower an individual for optimal response but induces unnecessary fear and worries [40,41,42]. In our study, patients who used a larger number of sources of information had higher odds of being worried, and patients who followed more credible sources of information had higher odds of being afraid during SARS-CoV-2 infection. The role of the media and public health communications must be understood and explored further as they will be an essential tool for combating health crises and future pandemics [43]. 

This study has several limitations. First, the range of experiences explored in this study is potentially limited by the sample—the response rate was intermediate and LTCF residents were not included. On the other hand, including the individuals in residential care could distort the results of the study as they had to cope with different challenges compared to the general population. Second, the questionnaire was addressed to cases with SARS-CoV-2 infection in June 2020, which may have led to recall bias, as in most the infection occurred in March and April 2020. Third, all data were self-reported and may have been susceptible to social desirability bias. It might be that some questions were not answered with honesty—the participants’ answers regarding isolation may have been more in line with the recommendations at the time than with their actual experience. On the other hand, a significant proportion stated that they did not isolate themselves from the rest of the household, which may speak in favor of sufficient openness of the participants.

## 5. Conclusions

Our data suggest that, when such a complex medical and social phenomenon as a pandemic occurs, it is necessary to take into account the possibility and ability of the individual to adapt to the situation and follow the recommendations, her/his emotional response and the stigma that accompanies new, emerging communicable diseases. Preparedness plans should also take into account these aspects in order to achieve an improved response to evolving health threats.

## Figures and Tables

**Figure 1 ijerph-19-12743-f001:**
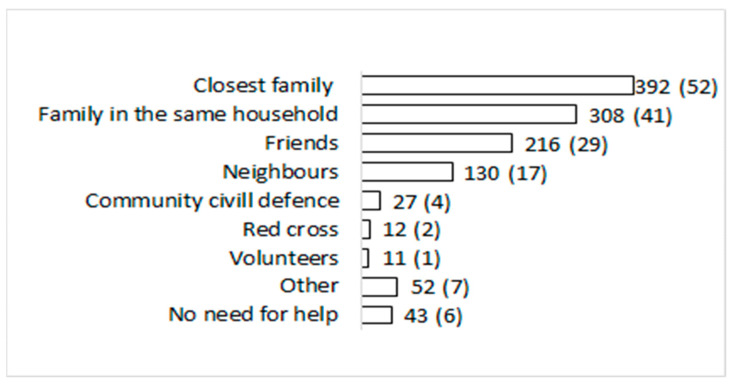
Caregivers and supporters for SARS-CoV-2 notified cases during isolation in the first pandemic wave; No. (%).

**Figure 2 ijerph-19-12743-f002:**
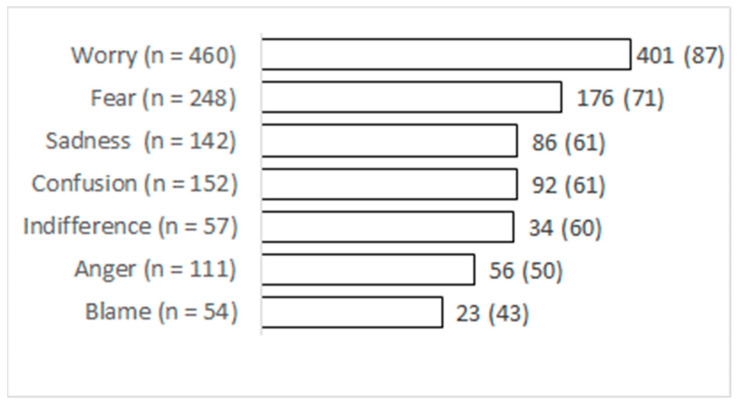
Emotional reactions that persisted from the time of diagnosis and during the acute phase of illness; No (%).

**Figure 3 ijerph-19-12743-f003:**
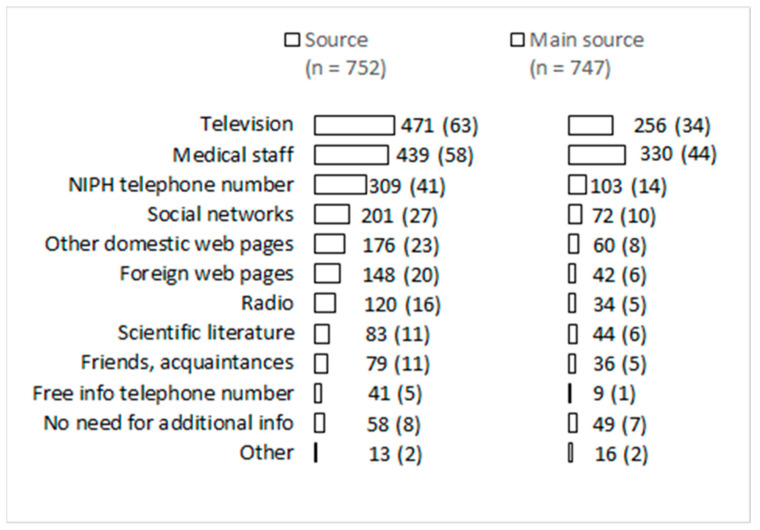
Source of information about COVID-19 for SARS-CoV-2 notified cases in the first pandemic wave; No (%) (NIPH = National Institute of Public Health).

**Table 1 ijerph-19-12743-t001:** Socio-demographic data and co-morbidities of SARS-CoV-2 notified cases in the first pandemic wave that responded to the survey.

Patients’ Data	No. (%)
Gender	
Female	418 (55.5)
Male	335 (44.5)
Median (IQR) age (n = 727)	49 (33–59)
Education	
Elementary	89 (11.9)
Secondary	327 (43.8)
University	331 (44.3)
Concomitant disease (n = 767)	381 (49.7)

**Table 2 ijerph-19-12743-t002:** The course of SARS-CoV-2 infection in the first wave—need for medical advice, hospitalization, symptoms on day 15 and ability to perform daily activities two weeks after the start of illness.

Course of SARS-CoV-2 Infection	No. (%)
Need for medical advice (n = 756)	503 (66.5)
Hospitalization (n = 767)	177 (23.1)
Median (IQR) duration of hospitalization (n = 175)	9 (5–16)
Symptoms on the 15th day (n = 760)	484 (63.7)
Everyday activities on the 15th day, the same as pre-COVID-19 (n = 746)	325 (43.6)

**Table 3 ijerph-19-12743-t003:** Support providers for SARS-CoV-2 notified cases in the first pandemic wave.

Support	No. (%)
Support from family members and friends	
No	4 (0.5)
Partly	27 (3.6)
Yes	716 (95.9)
Support from acquaintances and co-workers	
No	54 (7.4)
Partly	106 (14.5)
Yes	572 (78.1)
Support from medical staff	
No	47 (6.4)
Partly	70 (9.6)
Yes	615 (84)

**Table 4 ijerph-19-12743-t004:** Emotional reaction linked to infection with SARS-CoV-2 in notified cases in the first pandemic wave.

Feelings and Stigmatization	No. (%)
Feelings at diagnosis (n = 681)	
Sadness	145 (21.3)
Fear	256 (37.6)
Worry	479 (70.3)
Indifference	58 (8.5)
Anger	114 (16.7)
Blame	57 (8.4)
Confusion	165 (24.2)
Feelings during illness (n = 655)	
Sadness	120 (18.3)
Fear	215 (32.8)
Worry	452 (69)
Indifference	65 (9.9)
Anger	83 (12.7)
Blame	37 (5.6)
Confusion	121 (18.5)
Stigmatization	
No	391 (52.6)
Partly	186 (25)
Yes	167 (22.4)

**Table 5 ijerph-19-12743-t005:** Association between patient characteristics, support and source of information and the presence of negative feelings during SARS-CoV-2 acute infection in the first wave.

	Worry		Fear		Sadness	
	OR (95% CI)	*p*	OR (95% CI)	*p*	OR (95% CI)	*p*
Male gender	0.65 (0.44–0.95)	0.024	0.47 (0.32–0.69)	<0.001	0.74 (0.48–1.16)	0.188
Age	1.04 (1.03–1.05)	<0.001	1.02 (1.01–1.03)	<0.001	1.01 (1–1.02)	0.204
Having pet	1.09 (0.75–1.58)	0.669	0.81 (0.57–1.17)	0.264	0.81 (0.53–1.25)	0.349
SSS *	0.93 (0.51–1.69)	0.819	0.6 (0.34–1.04)	0.067	0.48 (0.27–0.88)	0.017
No isolation	0.92 (0.63–1.35)	0.676	0.87 (0.6–1.25)	0.446	0.59 (0.37–0.93)	0.022
NIS **	1.26 (1.12–1.43)	<0.001	1.05 (0.94–1.17)	0.401	1.03 (0.9–1.17)	0.693
NCS #	1.12 (0.82–1.52)	0.472	1.65 (1.22–2.21)	0.001	1.24 (0.88–1.76)	0.217

SSS * = strong social support (support from family, friends, colleagues and medical staff); NIS ** = number of sources of information; NCS # = number of most credible sources (phone number of the Ministry of Health, website of NIPH, medical staff and scientific literature).

## Data Availability

The data presented in this study are available upon request from the authors. The data are not publicly available.

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
