# Peer review of "Confronting SARS-CoV-2 Infection: Patients’ Experience in the First Pandemic Wave—Cross-Sectional Study"

_ijerph, 2022, doi:10.3390/ijerph191912743_

Round 1
Reviewer 1 Report
The research topic is very timely and worthy of publication after taking into account corrections and suggestions
Introduction
Please include more items of literature in the introduction. The phenomenon of Pandemic and its impact on various areas of life is widely described
Materials and Methods
1. Whether the subjects instructed or given verbal or written instructions on how to complete the questionnaire?
2.What measures have been taken to reduce possible errors in data collection?
3. Were and what were the selection and exclusion criteria used in the study? Was the only criterion for exclusion from the study the death of patients due to Covid-19.
4 Please attach to the manuscript a detailed graph showing the sample selection and the process of conducting the experiment.
5. please authors to include in the supplementary materials the questionnaire that was used to collect data
Results
1.line 109- In view of the fact of incomplete questionnaires, were these results taken for analysis?what criterion was used
2.Figure 3 and Table 4 - Please additionally provide an emotional response as a result of Covid-19 infection by gender. Have correlations been demonstrated and measured in this aspect. The gender of the subjects may condition the way they react and express their emotions, and certainly such an analysis would greatly enrich the presented research results.
4. Discussion
If the authors of the study have an analysis of the data including the gender of the subjects, it would be worth including references of this fact in the discussion
Author Response
Dear reviewer,
many thanks for your comments and suggestions.
Please find responses enclosed.
Introduction
Please include more items of literature in the introduction. The phenomenon of Pandemic and its impact on various areas of life is widely described.
We added more references in Introduction section to support the statement that pandemic brought many changes in everyday life (references from 6-10, following references re-numerated).
Materials and Methods
Whether the subjects instructed or given verbal or written instructions on how to complete the questionnaire?
The participants were given written instructions. The questions with more possible answers were marked and explained that multiple answers were possible.
What measures have been taken to reduce possible errors in data collection?
We kept the questionnaire as simple as possible to avoid misunderstanding the questions and to shorten the time needed to fill in the questionnaire. By doing this and including full information about the motivation for the research, we wanted to reduce the non-response bias. The data was collected via regular mail with the envelope ready for mailing the questionnaire back to the National Institute of Public Health. Simplification of the mailing procedure was therefore offered, which was aimed to increase the response rate. The chosen data collection method allowed also for the older (and other) people that are less IT literate to participate in the study and further reduced the possibility of non-response bias.
Were and what were the selection and exclusion criteria used in the study? Was the only criterion for exclusion from the study the death of patients due to Covid-19.
All first wave SARS-CoV-2 positive persons (symptomatic and asymptomatic) were included in the study with two exceptions: being a long-term care resident or we have already had the information that patient died within first 28 days after positive SARS-CoV-2 test. There were no other exclusion criteria (described in 2.1. Participants).
Please attach to the manuscript a detailed graph showing the sample selection and the process of conducting the experiment.
The graph showing the process of sample collection was included in the Supplement material section and a sentence added to the manuscript: The sample collection process was included in the Supplement material (Figure S1).
please authors to include in the supplementary materials the questionnaire that was used to collect data.
The questionnaire was translated to English and included in the Supplement section and a sentence added to the manuscript (2.2. Assessment Variables): The questionnaire is available in Supplement material section.
Results
line 109- In view of the fact of incomplete questionnaires, were these results taken for analysis?what criterion was used
Yes, all the answers per variable were taken into account – list wise deletion was not employed when describing each of the variables. The logistic regression models, however include only those respondents with complete data on all independent and each dependent variable.
Figure 3 and Table 4 - Please additionally provide an emotional response as a result of Covid-19 infection by gender. Have correlations been demonstrated and measured in this aspect. The gender of the subjects may condition the way they react and express their emotions, and certainly such an analysis would greatly enrich the presented research results.
We agree that emotional response could differ according to gender. That is why we included gender as a control variable in the logistic regression model (Table 5). From there it can be seen that men tended to experience worry and fear during the illness to lower extent than women. No difference was found in experiencing sadness, however. Maybe this was less clear as only “gender” was stated in Table 5 – now we added “Male gender”, so that it is clear that men had lower odds for experiencing worry, fear and sadness, but the latter was not statistically significant. We believe that Table 4 and Figure 2 are better left as they are – for the whole sample as the information on gender differences is provided in Table 5.
Discussion
If the authors of the study have an analysis of the data including the gender of the subjects, it would be worth including references of this fact in the discussion
We added the following sentence in Discussion section, 4.3. Feelings (including new references): In our study, male participants had lower odds for negative feelings such as worry and fear with no difference found regarding sadness compared to females. Female gender was reported frequently, but not consistently as risk factor for fear, anxiety and depression during first wave of COVID-19 pandemic [27,28].
Reviewer 2 Report
The article in very intesting.
The survey was done in very big sample - all positive cases in the country within three month period. This is very important for results and conclussion generalisation.
The inctrument was appropriate to assess the all variables.
Statistics used was appropriate - three multiple logistic regression models - to test association between demographic variables, sources of social support and information
The results are conclusive for prevention next pandemic wave stress in Slovenia and other countries.
Author Response
Dear reviewer,
Thank you very much for taking time to review our article
Reviewer 3 Report
The manuscript was well written. I have only one comment: change 2.1 'Patients" to 'Participants'
Author Response
Dear reviewer,
Thank you very much for taking the time to review our article.
We changed 2.1 Patients to 2.1.Participants.